# Antiviral Functions of Monoclonal Antibodies against Chikungunya Virus

**DOI:** 10.3390/v11040305

**Published:** 2019-03-28

**Authors:** Jing Jin, Graham Simmons

**Affiliations:** 1Vitalant Research Institute, San Francisco, CA 94118, USA; 2Department of Pathology and Laboratory Medicine, University of California, San Francisco, CA 94143, USA

**Keywords:** Chikungunya, antibodies, neutralization, budding, effector functions

## Abstract

Chikungunya virus (CHIKV) is the most common alphavirus infecting humans worldwide. Antibodies play pivotal roles in the immune response to infection. Increasingly, therapeutic antibodies are becoming important for protection from pathogen infection for which neither vaccine nor treatment is available, such as CHIKV infection. The new generation of ultra-potent and/or broadly cross-reactive monoclonal antibodies (mAbs) provides new opportunities for intervention. In the past decade, several potent human and mouse anti-CHIKV mAbs were isolated and demonstrated to be protective in vivo. Mechanistic studies of these mAbs suggest that mAbs exert multiple modes of action cooperatively. Better understanding of these antiviral mechanisms for mAbs will help to optimize mAb therapies.

## 1. Introduction

Chikungunya virus (CHIKV) is an emerging alphavirus. Spreading rapidly from endemic areas of Africa and Asia to Europe and the Americas, CHIKV is now the most common alphavirus infecting humans globally [1]. CHIKV is transmitted to humans by *Aedes* species mosquitoes and causes an acute febrile illness accompanied by severe arthralgia, with relapses lasting up to several months and recurring for several years in up to 40% of patients [2]. In a 2007 outbreak in India, the median workdays lost due to CHIKV infection was 35 [3]. This is one reason why CHIKV is classified by NIAID as a biodefense priority pathogen, despite low fatality rates. In a retrospective cohort study of the 2005–2006 La Réunion island outbreak, CHIKV was found to be a significant cause of central nervous system disease [4]. Currently, there are no vaccines or treatments for CHIKV infection. In the past decade, several potent human and mouse anti-CHIKV monoclonal antibodies (mAbs) were isolated and demonstrated to be protective in vivo [5,6,7,8,9,10,11,12,13]. In this review, we will discuss our understanding of the multiple antiviral mechanisms through which these mAbs exert viral inhibition.

## 2. Alphavirus Structure and Life Cycle

CHIKV is an enveloped, positive-stranded RNA virus belonging to the Alphavirus genus in the *Togaviridae* family. The CHIKV structural proteins, capsid (CP) and three envelope glycoproteins (GPs) (E1, E2 and E3), are synthesized as a polyprotein that is proteolytically processed. Cryo-electron microscopy (cryoEM) structure of mature alphavirus particles and virus-like particles (VLPs) revealed that virions are composed of two icosahedral layers: the outer envelope layer and the inner nucleocapsid (NC) core, both with T = 4 quasi-icosahedral symmetry. The envelope comprises 80 membrane embedded spikes, with each spike formed by a trimer of the E1–E2 heterodimer (Figure 1A) [13,14,15,16]. Each CP interacts with the cytosolic domain of E2, and 240 copies of CP form an icosahedral NC core enclosing the viral genomic RNA inside (Figure 1B,C). E1 is a type II membrane fusion protein and sits at the base of the trimeric spike, with E2 positioned on top of E1. E1 consists of three domains: domain I links distal domain II and membrane proximal domain III. A fusion loop is located at the distal end of E1 domain II. Crystal structures of CHIKV E1E2 heterodimer revealed three domains in E2 [17]: domain A is located in the center of the spike surface, while domain B and C are located at the distal end and the membrane proximal end of E2, respectively (Figure 1D). The apex of domain A and a part of domain B possess the putative binding site for receptor [13,18,19], which is confirmed by the recent identification of a shared receptor, Mxra8, for a number of arthritogenic alphaviruses including CHIKV, O’nyong nyong virus (ONNV), Mayaro virus (MAYV) and Ross River virus (RRV) [20]. At neutral pH, E2 domain B protects the fusion loop at the distal end of E1 from activation [17,21,22]. Cryo-EM structures of several alphaviruses have shown that E2 domain B has a lower electron density compared to other domains, suggesting that E2 domain B is flexible [13,15,16]. At low pH, E2 domain B is disordered and the fusion loop at the tip of E1 domain II is exposed in the X-ray crystal structure of envelope proteins from the related alphavirus, Sindbis [18].

In acidified endosomes, where alphaviruses are transported after endocytosis into target cells, low pH triggers conformational rearrangements in the envelope GPs to expose the fusion loop at the distal end of E1. Upon removal of E2 from the center of viral spike, E1 forms a homotrimer, further exposing the fusion loops of each monomer at the end of the trimeric complex for insertion into host membrane [23]. Membrane fusion between virus and host cell allows the penetration of nucleocapsids (NCs) into the cytosol and rapid disassembly. Viral nonstructural proteins are then translated from the incoming viral genomic RNA in the cytosol and form replication complexes to synthesize new viral RNA [24]. Viral structural proteins are translated from the newly synthesized viral subgenomic RNA in the cytosol. The positively charged residues at the N-terminal domain of CP interact with viral genomic RNA to nucleate the assembly of NCs [25,26]. GPs are synthesized as a polyprotein (p62-E1) containing E3E2 (p62) and E1. After co-translational proteolysis, the p62-E1 heterodimer then assembles into trimeric spikes in the endoplasmic reticulum and is delivered to the plasma membrane through intracellular membrane trafficking. Furin and furin-like host proteases cleave E3 during the transport through the *trans*-Golgi network to the plasma membrane, priming the spikes for fusogenic activation.

At the plasma membrane, the cytosolic domain of E2 binds to a hydrophobic pocket in CP on the surface of NC [27,28]. The icosahedral glycoprotein shell then assembles in synchrony with the icosahedral symmetry of the NC, finally resulting in budding of nascent virions. CPs can assemble into icosahedral NCs independently of GPs in situ and in vitro [29,30]. NCs can be prepared by in vitro assembly [30], isolation from virions stripped of envelope [31], or purification from infected cells [29]. Only a small subpopulation of these NCs were icosahedrally symmetric when visualized in vitro by cryoEM single particle analysis (SPA). The organization of CPs in the isolated NCs differed from that inside mature virions [29,30,31]. All these suggest that NCs are structurally heterogeneous and less stable than mature virions and undergo GP-driven and/or lipid-driven structural changes upon envelopment. CPs that are deficient in NC assembly were able to assemble and bud into wild-type-like Semliki Forest virus (SFV) virions driven by horizontal spike–spike interactions and vertical spike–capsid interactions [32,33], indicating a direct role for viral spikes in organizing the icosahedral conformation of alphavirus. In situ electron tomography of chemically fixed SFV-infected cells suggested that trimeric GP spikes might already assemble into hexagonal lattices in cellular membrane compartments prior to being delivered to the plasma membrane, where they are required for virus budding [34]. Mutations in the D-loop of the SFV E2 protein, that disrupt the contact between the viral E1 and E2 proteins, inhibit SFV budding, further suggesting that the native organization of the GP lattice is critical for virus budding [35]. Therefore, horizontal interactions between GPs and vertical interactions between GPs with CPs coordinately drive the formation of the icosahedral GP shell and induce membrane curvature formation required for virus budding, similar to COPI and COPII complex-driven membrane vesicle budding for intracellular protein transport [36].

## 3. Entry Neutralization by mAbs

Antiviral antibodies are traditionally screened for their ability to inhibit virus entry into target cells. Entry neutralization of envelope virus by mAbs can be mediated via two major mechanisms: inhibition of receptor binding and prevention of conformational changes within the viral fusion proteins post-receptor binding. For alphaviruses, membrane fusion occurs in acidified endosomes after virus enters the cells. In this review, neutralizing antibodies (NAbs) specifically refers to mAbs that inhibit virus entry into susceptible cells. The anti-CHIKV mAbs discussed below are summarized in Table 1.

### 3.1. Block Virus Binding to Receptors

Alanine scanning mutagenesis of E2 revealed the binding site of CHIKV receptor, the cell adhesion molecule Mxra8, on viral spikes: a solvent-accessible epitope spanning the top of E2 domain A and domain B [20]. A number of human mAbs targeting epitopes in E2 domain A (2H1, 8G18, 3E23 and 1O6) and human mAb 4J21 that binds epitopes spanning E2 domain A and domain B are able to inhibit binding of Mxra8 to CHIKV [10,20]. Among these mAbs, 4J21 binding to CHIKV VLPs was resolved by cryoEM. The footprint of 4J21 Fab covers primarily the β-ribbon connector that links E2 domain A and domain B together, as well as small regions on E2 domain A and E2 domain B (Figure 2) [37]. Cryo-EM structures of CHIKV VLP in complex with Fab fragments of human mAbs m242 and m10, as well as mouse mAb CHK-9, suggest these E2 domain A or domain B targeting mAbs block receptor binding to CHIKV (Figure 2) [13,43], although experiments are needed to demonstrate these mAbs prevent CHIKV from binding to Mxra8 and susceptible cells. In addition to blocking receptor binding [20], 4J21 is also able to inhibit low pH activated membrane fusion at the plasma membrane of susceptible cells in fusion-from-without (FFW) experiments, whereby after allowing virus to attach to cells at 4 °C, the environment is acidified in order to trigger viral and cellular membrane fusion at the cell surface [10]. In contrast, CHIKV VLPs in complex with Fab 8B10 aggregate at pH 6, suggesting that human mAb 8B10 is unable to prevent low pH activated membrane fusion while the cryo-EM structure of the complex suggests that 8B10 prevents virus binding to receptor [43]. One of the essential residues on E2 domain A for Mxra8 binding, W64, is a shared epitope residue for a number of human NAbs [7,10,20]. Escape mutants at this residue result in attenuated viral pathogenesis [7]. Human NAbs, C9 and IM-CKV063, target W64 on E2 domain A but do not prevent CHIKV from binding to Vero cells [7]. One possible explanation is that on Vero cells CHIKV can use alternative receptors/attachment factors that are independent of binding to W64. Alternatively, the 240 receptor binding sites on CHIKV are not sufficiently occupied by C9 or IM-CKV063 in order to block cell attachment, which we will discuss below.

### 3.2. Prevent Activation of Membrane Fusion

To prevent membrane fusion, NAbs need to keep E2 domain B in position to prevent exposure of fusion loop at low pH. In the cryoEM structures of CHIKV VLP in complex with Fab fragments of mouse mAb, CHK-152 [13] and human mAb, 5M16 [37], both mAbs span E2 domain A and domain B as well as the β-ribbon connector connecting the two domains within one E2 molecule (Figure 2). Both CHK-152 [8] and 5M16 [10] efficiently block CHIKV fusion at the plasma membrane in the FFW experiments. They may do so by cross-linking the flexible domain B to the rest of the virus surface thereby keeping the fusion loop unexposed beneath domain B. In fact, the density height of E2 domain B is similar to all the other domains in CHIKV VLP-Fab 5M16 [37]. In contrast, in the cryoEM structure of CHIKV VLP alone E2 domain B has lower density height compared to other domains [13]. This indicates stabilization of E2 domain B upon binding of 5M16 to CHIKV.

Human mAbs, C9 and IM-CKV063, isolated and characterized by our group and collaborators [5,9] neutralize CHIKV entry by inhibiting low pH-activated membrane fusion [7]. Both C9 and IM-CKV063 span domain A of one E2 molecule and domain B of the neighboring E2 molecule within one trimeric spike (Figure 3A,B), therefore stabilizing domain B to keep the fusion loop protected at low pH [7]. This is similar to the footprint of Fab 3B4C-4 on Venezuelan equine encephalitis virus (VEEV) [40]. As discussed above, although C9 and IM-CKV063 bind to W64 on E2 domain A that is essential for Mxra8 binding, they neutralize CHIKV entry by inhibiting membrane fusion not preventing virus binding to the target cell. It is likely that binding of only one Fab per trimeric spike is sufficient to prevent the conformational changes in GP required for activation of membrane fusion and hence lead to inhibition of entry. In contrast, to prevent receptor binding, all three receptor binding sites in each trimeric spike may need to be occupied. Probably steric hindrance prevents C9 and IM-CKV063 from occupying all three binding sites within one spike. This may partially explain how C9 and IM-CKV063 potently neutralize virus entry without inhibiting virus binding to the susceptible cell.

Broadly neutralizing antibodies (bNAbs) against HIV-1 [44], influenza virus [45], hepatitis C virus [46], Ebola virus [47], and dengue virus [48,49] have been isolated. These bNAbs are able to neutralize highly variable viruses by binding to conserved regions. CHIKV is relatively conserved genetically; with the envelope GPs sharing 95 to 99.9% identity across three major lineages (West African; East, Central, and South African; Asian). This explains neutralization of different strains across CHIKV lineages by NAbs isolated from recovered patients [5,10,12] or CHIKV-infected animals [8,50]. Interestingly, a panel of human and mouse bNAbs that neutralize multiple alphaviruses including CHIKV, MAYV, ONNV, RRV and SFV were identified [6]. Prophylactic treatment with two of these bNAbs protected mice against infection by CHIKV, MAYV and ONNV. These bNAbs target an epitope in E2 domain B that is conserved across multiple and distantly related arthritogenic alphaviruses. In the cryoEM structure of Fab fragment of one bNAb, CHK-265 in complex with CHIKV (Figure 3C), the Fab binds to E2 domain B and pushes it further down over the fusion loop. Concurrently, E2 domain A is moved around E1 domain II to the space between neighboring spikes (Figure 3D) [6]. This allows CHK-265 to crosslink E2 domain B with E2 domain A of the neighboring spike [6]. The electron density of E2 domain B is as high as the other GP domains, indicating the stabilization of E2 domain B explaining the mechanism for inhibition of membrane fusion by bNAb CHK-265.

Human mAb 5F10 also inhibits low pH-activated conformational changes by rigidly fixing E2 domain B, but without crosslinking domain B with stable domain A. Instead Fab 5F10 binds to domain B of each E2 molecule via CDRs and extends tangentially to the virus surface, leading to tight packing of Fabs bound on neighboring spikes [43]. As a result, the average density of E2 domain B are slightly higher than the densities of other domains and there is no measurable difference between the structures at pH 7.0 and pH 6.0, indicating the stabilization of domain B and inhibition of low pH activated conformational changes. Locking E2 domain B in position via packing Fabs bound on domain B of neighboring spikes was also reported for NAb 3B4C-4 against VEEV [40].

Epitopes for above-described anti-CHIKV NAbs are mostly complex quaternary epitopes, especially for ultra-potent multi-domain cross-linking NAbs, like 5M16, CHK-152 and IM-CKV063. This is reminiscent of potent NAbs against flaviviruses that target an epitope spanning three separate E protein monomers [51] or an epitope spanning both E proteins of an E protein dimer on mature virus [48]. It was reported that early neutralizing IgG responses to CHIKV in patients mostly targeted a single linear epitope (E2EP3) at the N-terminus of E2 [52]. E2EP3 conjugated to the large, multisubunit carrier protein Keyhole Limpet hemocyanin was reported to induce mild neutralizing activity and partial protection in vivo [52]. However, immunization with a pentamer of the E2EP3 peptide failed to induce NAbs and in vivo protection in mice [53]. In contrast, immunization with live wild-type or attenuated CHIKV [8,54,55,56,57], inactivated CHIKV [50], or VLPs [58], induces strong NAbs that provide in vivo protection in animals, probably due to better presentation of complex quaternary epitopes.

## 4. Budding Inhibition by mAbs

In addition to their canonical inhibition of virus entry into cells, human NAbs C9 and IM-CKV063 and mouse bNAbs CHK-265 and CHK-187 are able to inhibit virus release from CHIKV-infected cells [6,7]. Fab fragments of these NAbs do not inhibit virus release but retain the ability to bind their epitopes to neutralize virus entry [6,7]. Thus, the inhibition of virus release depends on bivalent binding of NAbs to CHIKV GPs. Resistance of escape mutants to NAb-mediated entry neutralization and budding inhibition indicates the same epitope residues are utilized by NAbs to neutralize mature virions and to inhibit virus budding on the cell surface [7]. It is unknown if NAbs bind to the same epitope in GPs presented on mature virions and on the surface of infected cells. Future studies to reveal the detailed structure of NAb crosslinking of GPs on cell surface are needed to address this question. Visualization of CHIKV-infected cells cultured in the presence of NAbs with STED super-resolution microscopy revealed large patches of CHIKV GPs bound with NAbs on the plasma membrane [41], suggesting that NAbs crosslink GPs and induce their coalescence into large patches on the plasma membrane. Thin-section transmission electron microscopy (TEM) imaging revealed a dense array of NAbs bound to GPs on the outer surface of CHIKV-infected cells and the inhibition of membrane curvature around NCs required for virus budding (Figure 4) [41]. In the cytosol, budding-arrested NC-like particles (NCLPs) varied in size and structure with no icosahedral symmetry detected by subtomogram averaging (STA) analysis of cryo-electron tomography (cryoET) of NAb-treated CHIKV-infected cells [41]. This suggests that inhibiting surface budding arrests intracellular NCs at assembly intermediates or forces NCs to disassemble. STA of membrane patches of NAb-induced coalesced GPs did not converge to any structure with symmetric organization of GPs, suggesting that they were conformationally and/or compositionally heterogeneous [41]. Therefore, C9 and IM-CKV063 inhibit CHIKV budding by disrupting assembly of GP spikes into an icosahedral lattice via crosslinking and coalescing viral GP spikes. Other than C9 and IM-CKV063, two above-mentioned human NAbs, 5F10 and 8B10, were also reported to strongly reduce extracellular virus released from CHIKV-infected cells [12], although study is needed to demonstrate if they inhibit virus release at the budding step. The budding inhibition mechanism of antiviral function of mAbs is probably independent of the ability to neutralize virus entry by mAbs. A mAb CK47 targeting CHIKV E1 domain III is able to inhibit virus release but not entry [42]. One mAb targeting E3 of VEEV was reported to inhibit virus release from VEEV-infected cells but not virus entry into target cells [59]. Further studies are needed to understand the mechanism behind the inhibition of virus release by these non-neutralizing mAbs. As most of the mAbs screening focused on entry neutralization, unbiased screening of mAbs for budding inhibition function is required to isolate budding inhibitory mAbs and test their in vivo protection efficiencies. Studies of budding inhibitory mAbs targeting different epitopes will help us better understand the mechanism behind mAb-mediated inhibition of alphavirus budding.

The phenotype that mAbs inhibit virus release was observed for viruses from a number of families including influenza virus (IAV) [60,61], bovine leukemia virus [62], herpes virus [63], vaccinia virus [64], rubella virus [65], and Marburg virus (MARV) [66]. Most of those studies utilized mAbs or anti-sera that do not independently neutralize virus entry and did not resolve the mechanisms underlying virus release inhibition. Thin-section TEM studies with two non-NAbs against MARV [66] and a broadly reactive mAb against IAV [61] suggested that those mAbs tether budded virions to viral GPs on the cell surface. The budding inhibition function of anti-CHIKV mAbs described in this section is clearly distinct from the tethering mechanism for mAbs against MARV and IAV. It is probably unique for alphavirus as lateral organization of viral spikes into an icosahedral lattice drives alphavirus assembly/budding on the cell surface; therefore, alphavirus budding is targeted by humoral immunity. 

## 5. Antibody-Activated Effector Functions

Through interaction with Fc receptors expressed on all innate immune cells, mAbs mediate an array of effector functions that are important contributors to protective immunity [67]. These innate immune effector functions include antibody-dependent, cell-mediated phagocytosis (ADCP) of virions and infected cells by monocytes, macrophages and neutrophils; direct killing of infected cells via antibody-dependent, cellular cytotoxicity (ADCC) by NK cells and complement activation [68]. Mounting evidence suggests that Fc-effector functions elicited by mAbs play a critical role in protection against infection [69,70,71,72]. Humans have six classical Fc receptors (FcγRI, FcγRIIa, FcγRIIb, FcγRIIc, FcγRIIIa and FcγRIIIb) expressed in varied combinations on all innate immune cells at different levels [73]. Of the six FcγRs, FcγRIIb is the sole inhibitory receptor. The cellular outcome of IgG-FcγR interactions is determined by the affinity of the Fc for the specific receptor and the expression pattern of FcγRs on effector cells [74]. Classical FcγRs bind close to the hinge region of IgG [75,76,77] and activating receptors signal through an immunoreceptor tyrosine-based activation motif (ITAM) either in the cytoplasmic domain or on the common gamma chain that clusters during activation, initiating signaling cascades that activate immune cells to induce effector functions [78]. Except for FcγRI, all the other FcγRs exhibit low affinity for IgG and therefore are unable to bind monomeric IgG [67]. IgG-antigen complexes are able to sufficiently engage and crosslink low-affinity FcγRs via multi-valent, high avidity interactions leading to signaling from oligomerized FcγRs [79,80]. As described above, NAbs C9 and IM-CKV063 induced coalescence of viral GP-NAb complexes on the plasma membrane of CHIKV-infected cells leading to inhibition of virus budding [41]. The dense layer of coalesced NAb-GP complexes on the surface of CHIKV-infected cells could present multi-valent Fc to FcγRs and potently activate ADCC from effector cells that were engineered to express the activating FcγR, FcγRIIIa [41]. This is supported by the finding that bivalent binding of NAbs to viral GPs is required for the potent activation of ADCC (data unpublished). Overall these results suggest that motility of antigen on the plasma membrane induced by bivalent binding of IgGs contributes to potent ADCC activation. An assay targeting virus-infected cells is better suited to detect this ADCC activation mechanism compared to assays based on antigen immobilized on plates [72,81,82,83].

Very recently, Fox et al reported that optimal therapeutic activity of mAbs against CHIKV requires Fc-FcγR interaction on monocytes [38]. Humanized CHK-152 and CHK-166 of human IgG1 subclass, but not their N297Q mutants that do not interact with FcγR and C1q, were able to promote immune cell infiltration, viral clearance and reduce mouse foot swelling when administrated at 3 days post-infection (about the time of peak viremia) [38]. Similar protection by these mAbs was observed in mice lacking C1q as in wild type mice, but not in mice lacking the Fc receptor common γ chain that abrogates expression of activating FcγRs, suggesting that Fc-FcγR interactions, rather than complement, mediate the clinical and virological protection [38]. Ex vivo phagocytosis experiments with CHIKV p62-E1-coated beads demonstrated ADCP by primary murine monocytes and neutrophils [38]. For GP-icosahedral alphaviruses, the density and quaternary structure of GPs on virions and the surface of infected cells may be different from that of the recombinant protein coated beads. It will be interesting to test ADCP of native virions and virus-infected cells. Very importantly, in this study, the mAb-mediated protection was demonstrated to depend on monocytes, not neutrophils or NK cells, by experiments with mice depleted of individual cell types [38]. Because the NK cell is the only innate immune cell solely expressing activating FcγR, RcγIIIa, dogma has dictated that NK cells are the major mediators of ADCC in humans [84]. Using FcγR-humanized mice that express the full array of hFcγRs on a background lacking all murine FcγRs [85], DiLillo and Ravetch have demonstrated that macrophages mediate ADCC of antibody-coated target cells in vivo in the context of huIgG1 antibody and the human FcγR system [86]. It will be interesting to test primary monocyte/macrophage mediated ADCC of mAb-coated CHIKV-infected cells and determine FcγRs expressed on monocyte/macrophage that signal the effector functions. Optimizing the interactions between Fc and specific FcγRs will further improve in vivo protection by mAbs [87,88,89]. 

## 6. Neutralizing IgM mAbs

The anti-CHIKV mAbs described so far are all IgGs that represent approximately 75% of serum antibodies in humans. Before the IgG response, induction of IgM confers protection against infection of West Nile virus [90], rabies virus [91], influenza virus [92], smallpox virus [93] and related alphaviruses SFV [94] and Sindbis [95]. A potent neutralizing IgM against CHIKV, 3E7b, was reported [11]. 3E7b targets N218 on E2 domain B and blocks virus binding to susceptible cells [11]. It potently neutralizes CHIKV entry with an IC_50_ of 4.5 ng/mL [11], comparable with that of the strongest human neutralizing IgG, 5M16, 3.4 ng/mL [37]. The pentameric structure of IgM may render it potent for crosslinking and coalescing GPs on CHIKV-infected cells, therefore strongly inhibit virus budding. In addition, IgM is often considered to be the most potent immunoglobulin in complement activation through the classical pathway [96]. It will be interesting to test if neutralizing IgMs can inhibit CHIKV budding and activate the complement response to clear virus-infected cells. 

## 7. Development of Antibody Therapeutics for CHIKV-Infection

Treatment with a number of mAbs in immunocomprised animal models, neonatal mice [9], *Ifnαr*−/− mice [5,8,10], or AG129 mice [39], protects against lethal CHIKV challenge when administrated 24 hours before, or up to 60 hours post-infection. When administered as post-exposure therapy, a number of mAbs limited viral spread to joints and muscles and reduced tissue inflammation in wild-type mice [7,8,38,97] and in rhesus macaques [8,97,98]. Acute CHIKV infection in mice or rhesus macaques induces viremia that lasts for approximately 3–4 days, with peak viral titers detected at 2- or 3-days post-infection [50,99,100]. In humans the acute viremic phase of CHIKV-infection lasts 4–12 days post-symptom onset [101]. High CHIKV viremia is associated with increased severity of illness [99,102]. Treatment with anti-CHIKV mAbs in animal models during the viremic phase successfully reduces virus dissemination to the distal joints and tissues [38,97,98]. However, the treatment window is narrow. Future studies are needed to test if so-far identified protective mAbs are able to reduce tissue inflammation or prevent CHIKV persistence when administrated post-viremia. Developing better animal models that fully reproduce the chronic rheumatoid syndrome following CHIKF is urgently needed.

The high mutation rate of alphviruses helps virus to escape from mAb therapeutics, especially when administrated during the acute phase because the high viral load increases the chance for pre-existing or selected resistant variants to emerge. Combination of mAbs that target different antigenic sites, eg. CHK-152 and CHK-166, limits the selection of escape mutants and provides efficient protection [8,98]. Alternatively, if mAbs target an epitope that is critical for virus growth, fitness and viral pathogenesis, escaping mutation will be lethal for the virus or attenuate viral pathogenesis. Under these conditions, single mAb treatment is able to protect the host effectively. For example, the escape mutant to IM-CKV063 has a single W to G mutation at amino acid 64 of E2 domain A that is critical for receptor Mxra8 binding [7]. W64G mutant CHIKV fails to cause death in neonatal mice and results in less severe joint disease compared to wild-type CHIKV [7]. IM-CKV063 administrated post-infection provides superior protection against CHIKV-induced arthritis in 3-weeks-old mice compared to a panel of NAbs including CHK-152 and CHK-166 [7].

In recent years, technological advances in high-throughput B cell isolation, transgenic animal strains hosting human immunoglobulin sequences, antibody display platforms and antibody engineering have drastically accelerated the discovery and optimization of antiviral mAbs. Many excellent reviews on these new technologies have been published recently [103,104]. With advanced technologies, more highly potent and broadly cross-reactive human mAbs will soon be added to our current arsenal to fight CHIKV and other alphaviruses.

## 8. Outstanding Questions and Conclusions

Traditionally, antiviral antibodies are screened for their abilities to neutralize virus entry in vitro and extensively studied for their neutralizing mechanisms, including inhibition of receptor binding and prevention of conformational changes within fusion proteins [105]. In vivo protection conferred by non-NAbs was reported for H7N9 avian influenza virus [106], HIV [107] and Ebola virus [108,109]. In a recent systematic analysis of mAbs against Ebola virus GP, immune effector functions were found to contribute to, and may drive, mAb-mediated protection [72,83]. Considering the multiple array of antiviral mechanisms for mAbs against CHIKV reviewed here: entry neutralization, budding inhibition and potent ADCC/ADCP activation, we propose a “multi-functional” model for mAbs against CHIKV (Figure 5). To protect host from CHIKV infection, mAbs exert multiple mechanisms of action synergistically in vivo. A single neutralization assay is not always predictive of in vivo efficacy of mAbs. Comprehensive screening covering the entire viral life cycle and reproducing the diversity of Fc receptors and effector cells are needed to identify best protective mAbs. What the contribution of each of these antiviral mechanisms is to in vivo protection and how different antiviral mechanisms cooperate spatially and temporally in the host warrant future studies.

## Figures and Tables

**Figure 1 viruses-11-00305-f001:**
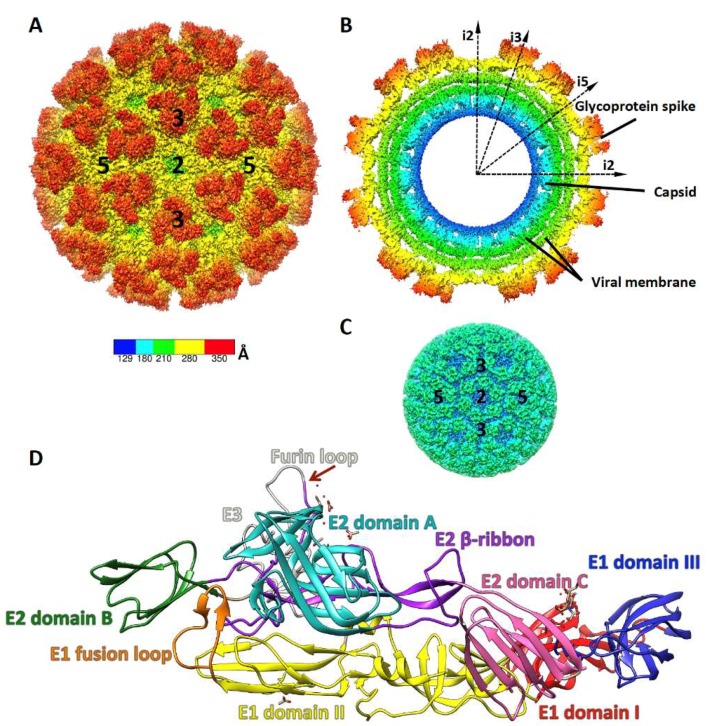
Structure of Chikungunya virus (CHIKV) and envelope glycoproteins. CryoEM density map (EMD-5577) of (**A**) CHIK virus-like particles (VLP), (**B**) CHIK VLP viewed at the cross-section, and (**C**) CHIK nucleocapsid core, colored according to the radial distance from the center of the virus; (**D**) Structure of the p62-E1 heterodimer (PDB:3N40).

**Figure 2 viruses-11-00305-f002:**
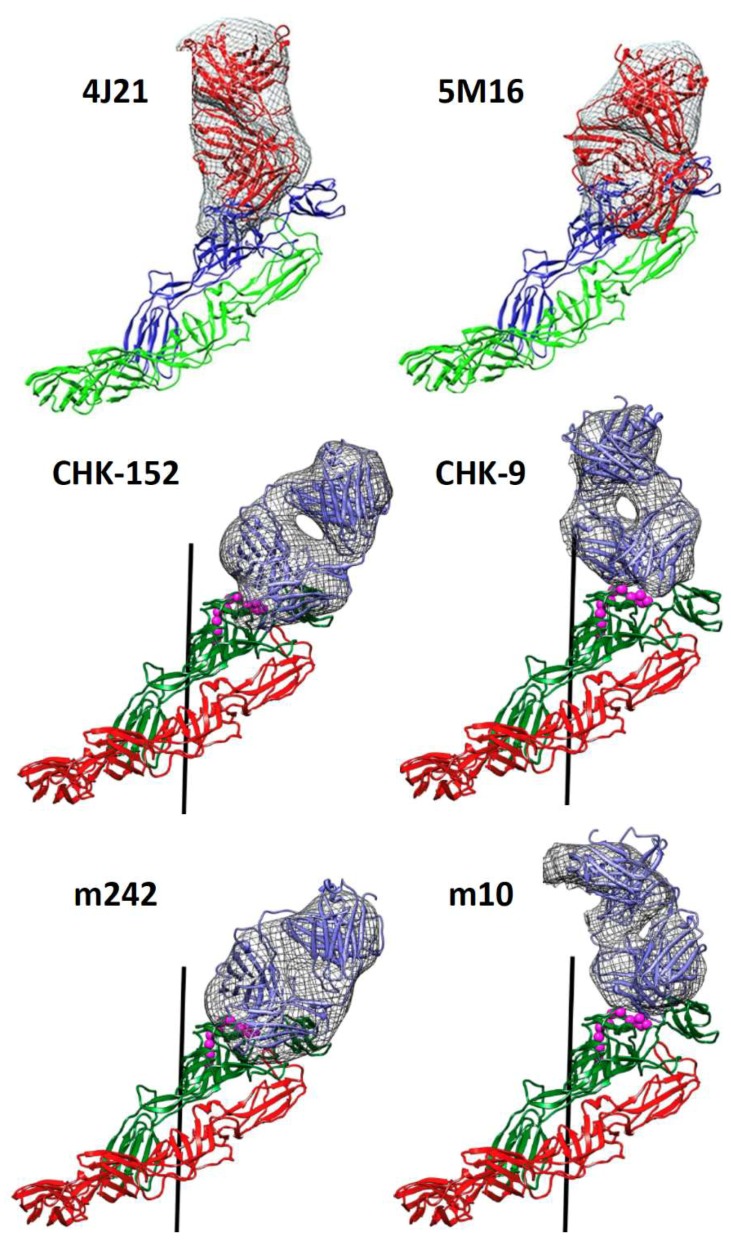
Fit of Fab structures into the cryoEM density “difference” maps (grey mesh surface) bound to E1-E2 heterodimer. Binding of 4J21 and 5M16 Fabs to E1 (green)-E2 (blue) heterodimer was adapted from [37]. The Fab structures are shown in red and fitted into the cryo-EM density. E1 is shown in green, and E2 is shown in blue. Binding of CHK-152, CHK-9, m242 and m10 Fabs to E1 (red)-E2 (green) heterodimer was adapted from [13]. The Fab structures are shown in blue and fitted into the cryo-EM density.

**Figure 3 viruses-11-00305-f003:**
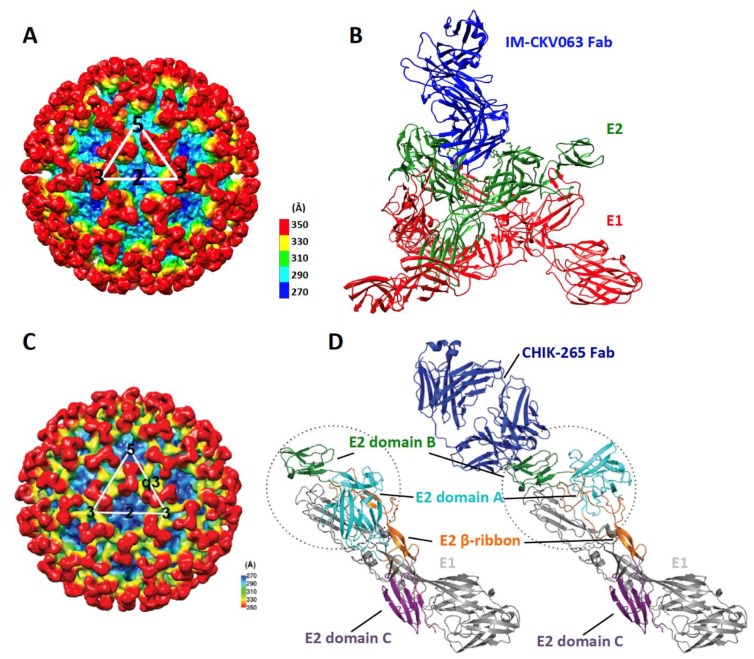
Binding of IM-CKV063 and CHK-265 to CHIKV. (**A**) CryoEM reconstruction of CHIK VLP in complex with IM-CKV063 Fab fragment (EMD-6457); (**B**) Structure of the trimeric spike bound with IM-CKV063 Fab as observed in (**A**), showing IM-CKV063 crosslinks two neighboring E2 molecules within one trimeric spike. E1, E2 and Fab are shown in red, green and blue, respectively. (**C**) CryoEM reconstruction of CHIK vaccine strain 181/25 in complex with CHIK-265 Fab fragment; (**D**) (Left) Structure of the E1-E2 heterodimer (PDB:3N42); (Right) Structure of the E1-E2 heterodimer bound with CHIK-265 Fab as observed in (**C**), showing the repositioning of E2 domain A upon CHIK-265 binding to E2 domain B. The E2 domain A and domain B are circled, showing the difference in their conformations with and without Fab CHIK-265 binding (adapted from [6] with permission).

**Figure 4 viruses-11-00305-f004:**
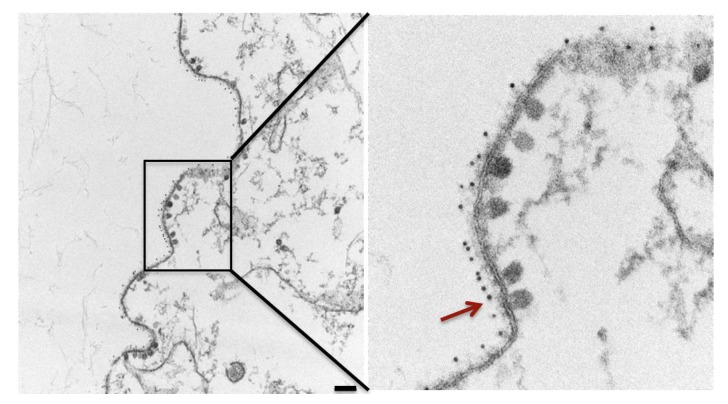
Neutralizing antibodies act on the outer leaflet of the plasma membrane to block CHIKV budding. CHIKV-infected cells treated with C9 were labeled with gold-conjugated anti-human antibodies before visualized by thin-section TEM. Budding arrested NCLPs are indicated with a red arrow. Scale bar: 100 nm (adapted from [41] with permission).

**Figure 5 viruses-11-00305-f005:**
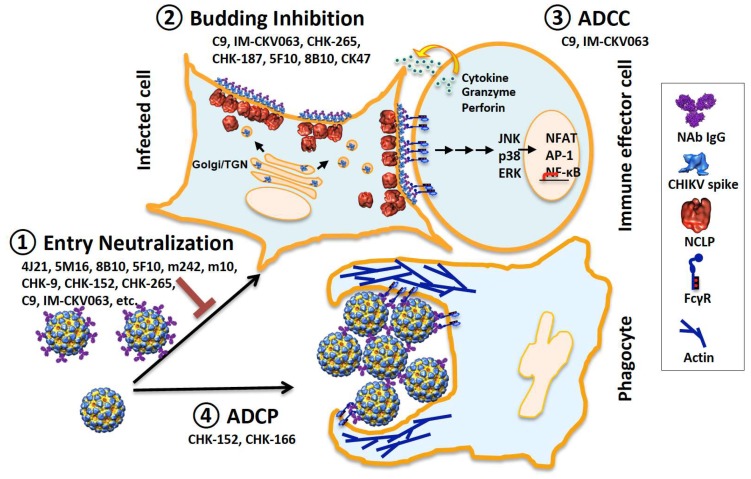
Model of multi-antiviral mechanisms for mAbs against CHIKV. Anti-CHIKV mAbs inhibit virus spreading by neutralizing virus entry into susceptible cells (**1**), inhibiting nascent virus budding from CHIKV-infected cells (**2**), activating ADCC to clear CHIKV-infected cells (**3**) and ADCP to clear virions (**4**). NCLP, nucleocapsid like particle; ADCC, antibody-dependent, cellular cytotoxicity; ADCP, antibody-dependent, cell-mediated phagocytosis.

**Table 1 viruses-11-00305-t001:** Characteristics of mAbs against CHIKV.

mAb	Species	Antigenic Sites	CHIKV-LR EC_50_ ng/mL (CI)	Demonstrated Mode of Action	Reference
2H1	human	E2-DA	5.6 (4.9–6.3)	entry neutralization	[10]
8G18	human	E2-DA	7.3 (6.3–8.4)	entry neutralization	[10]
3E23	human	E2-DA	18.7 (16.3–21.5)	entry neutralization	[10]
1O6	human	E2-DA	57.5 (48.8–68.1)	entry neutralization	[10]
4J21	human	E2-DA, DB, arch	7.4 (6.6–8.3)	entry neutralization	[10,37]
5M16	human	E2-DA, DB, arch	5.9 (5.0–6.8)	entry neutralization	[10,37]
CHK-152	mouse	E2-DA, DB, arch	5 (5–7)	entry neutralization, ADCP	[8,12,13,38,39,40]
CHK-9	mouse	E2-DA	36 (31–43)	entry neutralization	[13,8]
CHK-166	mouse	E1-DII	154 (116–205)	entry neutralization, ADCP	[8,12,38,39,40]
CHK-265	mouse	E2-DA, DB	8 (7–9)	entry neutralization, viral egress inhibition	[6,8]
CHK-187	mouse	E2-DB	11 (9–12)	entry neutralization, viral egress inhibition	[6,8]
C9	human	E2-DA, DB, arch	51 *	entry neutralization, budding inhibition, ADCC	[7,9,41]
IM-CKV063	human	E2-DA, DB, arch	7.4 *	entry neutralization, budding inhibition, ADCC	[5,7,41]
m242	human	E2-DA	300	entry neutralization	[13]
m10	human	E2-DB	7	entry neutralization	[13]
8B10	human	E2-DA, DB	46 ^#^	entry neutralization, budding inhibition	[12,39,40]
5F10	human	E2-DB	62 ^#^	entry neutralization, budding inhibition	[12,39,40]
CK47	mouse	E1-DIII	no	viral egress inhibition	[42]

Neutralizing activity was determined by FRNT on Vero cells with serial dilutions of mAbs and 100 FFU of the CHIKV-LR strain. * neutralization of reporter HIV pseudotyped with CHIKV S27 E1/E2. ^#^ neutralization of CHIKV11 strain by PRNT assay.

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
