# Peer review of "Antiviral Functions of Monoclonal Antibodies against Chikungunya Virus"

_viruses, 2019, doi:10.3390/v11040305_

Round 1
Reviewer 1 Report
Jin and Simmons have put up a nice paper that describes the activities of antibodies against chikungunya viruses. The paper explains the different functions and mechanisms of the antibodies nicely, describing specifically the structural changes to the virus due to the interactions with the antibodies. It is easy to read and provides a comprehensive overview of how antibodies function, which can be also applied to the studies of other infectious agents.
Comments:
1. It would be good if the authors can further elaborate on the topic of cross-reactivity, especially between CHIKV and ONNV (as well as other alphaviruses). Stating clearly their structural similarities and how antibodies raised against one virus can affect and protect against the other virus. This will be important, as there are some outbreaks of ONNV ongoing.
2. In terms of antibodies being elicited in response to chikungunya, some of these antibodies target linear epitopes. Could the authors also comment on the feasibility or non-feasibility on producing monoclonal antibodies that target linear epitopes versus those that target conformational antibodies? Is it worthwhile to develope linear-epitopes-recognizing monoclonal antibodies?
3. Also, I feel that the paper lacks a section that describes the future of monoclonal antibodies in infections. The authors should include this section and state clearly the upcoming technologies, challenges and opportunities in this area of research. What about the development of chimeric humanized antibodies? And also the usage of transgenic animals in producing antibodies that are more “human-like?”
Author Response
Jin and Simmons have put up a nice paper that describes the activities of antibodies against chikungunya viruses. The paper explains the different functions and mechanisms of the antibodies nicely, describing specifically the structural changes to the virus due to the interactions with the antibodies. It is easy to read and provides a comprehensive overview of how antibodies function, which can be also applied to the studies of other infectious agents.
Comments:
1. It would be good if the authors can further elaborate on the topic of cross-reactivity, especially between CHIKV and ONNV (as well as other alphaviruses). Stating clearly their structural similarities and how antibodies raised against one virus can affect and protect against the other virus. This will be important, as there are some outbreaks of ONNV ongoing.
The mechanism of bNAbs against multiple alphaviruses was explained as “These bNAbs target a conserved epitope in E2 domain B.” In the revised version, in page 12 line 7-8, it was changed to “These bNAbs target an epitope in E2 domain B that is conserved across multiple and distantly related arthritogenic alphaviruses.”
“Prophylactic treatment with two of these bNAbs protected mice against infection by CHIKV, MAYV and ONNV” was added in page 12 line 6-7.
2. In terms of antibodies being elicited in response to chikungunya, some of these antibodies target linear epitopes. Could the authors also comment on the feasibility or non-feasibility on producing monoclonal antibodies that target linear epitopes versus those that target conformational antibodies? Is it worthwhile to develope linear-epitopes-recognizing monoclonal antibodies?
Divergent opinion on induction of protective antibodies by linear epitope is now discussed from page 12, line 25 to page 13, line 6.
3. Also, I feel that the paper lacks a section that describes the future of monoclonal antibodies in infections. The authors should include this section and state clearly the upcoming technologies, challenges and opportunities in this area of research. What about the development of chimeric humanized antibodies? And also the usage of transgenic animals in producing antibodies that are more “human-like?”
These are now briefly discussed in page 18, lines 22-28, to keep this review focused on different antiviral mechanisms for anti-CHIKV mAbs. There are many excellent reviews on antibody discovery and engineering published in recent years. References for two selected reviewers were included in the revised manuscript.
Reviewer 2 Report
In this review, Jin and Simmons describe the isolation and characterization of several chikungunya virus (CHIKV)-specific monoclonal antibodies by different investigators in the field. The manuscript focuses on three major antiviral modes of action for these antibodies, including blockade of virus entry, inhibition of virus budding, and induction of effector functions in innate immune cell subsets (e.g. antibody-dependent cellular cytotoxicity). For the mechanisms of direct virus neutralization, the authors detail the predominant structural basis for antibody-mediated inhibition using published structures and models of Fab-glycoprotein interactions. The authors go on to describe work that has been done to understand indirect inhibition of virus infection through activation of innate immune cell functions, specifically focusing on studies of Fc receptors and complement in vitro and in vivo. In general, this review presents a thorough summary of the work to develop CHIKV-specific monoclonal antibodies and understand mechanisms of virus neutralization. While this review provides a valuable and much needed resource for the field, expanding upon the topics discussed and including a few unaddressed areas as detailed below would greatly enhance the impact of the manuscript.
General comments:
1. Although this review comprehensively describes the specific antibodies identified and characterized by the field, it is lacking any discussion of the therapeutic potential for these antibodies. In particular, additional text should be included near the end of the review that discusses the following unaddressed areas:
a. Whether blockade at specific steps in the virus replication cycle or certain antibody-elicited antiviral functions are more efficacious than others for protection. Which have the greatest therapeutic potential?
b. The efficacy of these mAbs in in vivomodels of infection. What is the therapeutic window in which these antibodies are effective?
c. Strain specificity of the discussed antibodies. Which epitopes should be targeted to provide the broadest protection? Are all strains expected to be equally susceptible to blockade at different steps in the virus replication cycle?
d. Viral escape mutations identified in the mentioned studies and whether antibody combinations should be used for effective treatment.
2. The authors discuss broadly neutralizing antibodies (bNabs) for CHIKV, but do not describe which characteristics make these particular antibodies broadly neutralizing. Do these bNabs target different or unique epitopes? If different, are the targeted epitopes involved in different or the same phases of viral replication? Are these epitopes conserved across different CHIKV strains or even alphaviruses? It is important to answer these questions to understand the therapeutic potential of these antibodies.
3. Given the breadth of antibodies discussed in this review, the addition of a table early in the review that outlines pertinent details for each antibody (e.g. the species, IC50, mechanism of action) would be a helpful reference for readers. Beginning with a tabular summary would help put the reviewed studies in context to one another before expounding upon structural and mechanistic specifics for each antibody. This will help with clarity and comprehension of the review.
4. At times, the text was very cumbersome to read. Many sentences (e.g. lines 87-90 and 230-232) were very long and difficult to follow. Some simple editing throughout to make the text more precise and concise would greatly improve the readability. Breaking down long sentences covering several thoughts into two or more sentences would be very helpful.
Specific comments:
1. In the section titled “Budding Inhibition by mAbs”, the authors refer to mAbs that block virus budding as “non-neutralizing” (line 212), but the word “neutralization” was never clearly defined by the authors. As written, the sentence implies that blocking virus budding is not an efficacious antiviral mechanism. If “non-neutralizing” in this context actually means non-entry blocking, the authors should make this correction. Perhaps it would be best to clearly define at the start what is meant by “neutralizing”.
2. Fig. 1D: The structure and labels are too small to see clearly.
3. Line 37: please define GPs.
4. Line 115: 8B10 is mentioned for Fig. 2, but is not included in the figure. This antibody should be mentioned separately.
5. Fig. 2: The images should be increased in size to see differences between the antibody structures. Also, labels or color descriptions should be included in the figure or legend. It would also help to discuss the antibodies in the order that the structures are presented in the figure.
6. Fig. 3A: The numbering on the legend is too small to see clearly.
7. Fig. 3D&E: Labels or color descriptions should be included in the figure or legend as well as asterisks or arrows to indicate sites referenced in the text.
8. Fig. 5: It could be useful to include antibodies mentioned in the text under their respective mechanisms in this figure.
9. Lines 239-240: “infection by mAbs” is confusing. The sentence should be reworded, such as “Mounting evidence suggests that Fc-effector functions elicited by mAbs play a critical role in protection against infection.”
Author Response
In this review, Jin and Simmons describe the isolation and characterization of several chikungunya virus (CHIKV)-specific monoclonal antibodies by different investigators in the field. The manuscript focuses on three major antiviral modes of action for these antibodies, including blockade of virus entry, inhibition of virus budding, and induction of effector functions in innate immune cell subsets (e.g. antibody-dependent cellular cytotoxicity). For the mechanisms of direct virus neutralization, the authors detail the predominant structural basis for antibody-mediated inhibition using published structures and models of Fab-glycoprotein interactions. The authors go on to describe work that has been done to understand indirect inhibition of virus infection through activation of innate immune cell functions, specifically focusing on studies of Fc receptors and complement in vitro and in vivo. In general, this review presents a thorough summary of the work to develop CHIKV-specific monoclonal antibodies and understand mechanisms of virus neutralization. While this review provides a valuable and much needed resource for the field, expanding upon the topics discussed and including a few unaddressed areas as detailed below would greatly enhance the impact of the manuscript.
General comments:
1. Although this review comprehensively describes the specific antibodies identified and characterized by the field, it is lacking any discussion of the therapeutic potential for these antibodies. In particular, additional text should be included near the end of the review that discusses the following unaddressed areas:
a. Whether blockade at specific steps in the virus replication cycle or certain antibody-elicited antiviral functions are more efficacious than others for protection. Which have the greatest therapeutic potential?
The discovery of the multi-mode antiviral mechanisms for anti-CHIKV mAbs occurred recently. What the contribution of each mechanism is to protection is an open question and warrants future studies (see page 19, lines 14-16).
b. The efficacy of these mAbs in in vivo models of infection. What is the therapeutic window in which these antibodies are effective?
This is discussed in Section 6 added to the revised manuscript from page 17 line 22.
c. Strain specificity of the discussed antibodies. Which epitopes should be targeted to provide the broadest protection? Are all strains expected to be equally susceptible to blockade at different steps in the virus replication cycle?
From page 11, line 4 to page 12, line 3, neutralization of different CHIKV strains was discussed. Page 13, lines 14-19 discussed the epitopes for entry neutralization and budding inhibition by NAbs.
d. Viral escape mutations identified in the mentioned studies and whether antibody combinations should be used for effective treatment.
This is discussed in Section 6 added to the revised manuscript from page 18 line 8.
2. The authors discuss broadly neutralizing antibodies (bNabs) for CHIKV, but do not describe which characteristics make these particular antibodies broadly neutralizing. Do these bNabs target different or unique epitopes? If different, are the targeted epitopes involved in different or the same phases of viral replication? Are these epitopes conserved across different CHIKV strains or even alphaviruses? It is important to answer these questions to understand the therapeutic potential of these antibodies.
The mechanism of bNAbs against multiple alphaviruses was explained as “These bNAbs target a conserved epitope in E2 domain B.” In the revised version, in page 12 line 7-8, it was changed to “These bNAbs target an epitope in E2 domain B that is conserved across multiple and distantly related arthritogenic alphaviruses.”
1. Given the breadth of antibodies discussed in this review, the addition of a table early in the review that outlines pertinent details for each antibody (e.g. the species, IC50, mechanism of action) would be a helpful reference for readers. Beginning with a tabular summary would help put the reviewed studies in context to one another before expounding upon structural and mechanistic specifics for each antibody. This will help with clarity and comprehension of the review.
Table 1 with all discussed anti-CHIKV mAbs was added in the revised manuscript.
4. At times, the text was very cumbersome to read. Many sentences (e.g. lines 87-90 and 230-232) were very long and difficult to follow. Some simple editing throughout to make the text more precise and concise would greatly improve the readability. Breaking down long sentences covering several thoughts into two or more sentences would be very helpful.
Long sentences were broken down if possible.
Specific comments:
1. In the section titled “Budding Inhibition by mAbs”, the authors refer to mAbs that block virus budding as “non-neutralizing” (line 212), but the word “neutralization” was never clearly defined by the authors. As written, the sentence implies that blocking virus budding is not an efficacious antiviral mechanism. If “non-neutralizing” in this context actually means non-entry blocking, the authors should make this correction. Perhaps it would be best to clearly define at the start what is meant by “neutralizing”.
In page 6, lines 17-19, “In this review, neutralizing antibodies (NAbs) specifically refers to mAbs that inhibit virus entry into susceptible cells” was added to clarify.
2. Fig. 1D: The structure and labels are too small to see clearly.
Fig. 1D was reproduced with larger labels.
2. Line 37: please define GPs.
In page 3, line 20, “GPs” was replaced with “glycoproteins (GPs)”.
4. Line 115: 8B10 is mentioned for Fig. 2, but is not included in the figure. This antibody should be mentioned separately.
In revised manuscript, 8B10 was discussed separately in page 8, lines 2-5.
5. Fig. 2: The images should be increased in size to see differences between the antibody structures. Also, labels or color descriptions should be included in the figure or legend. It would also help to discuss the antibodies in the order that the structures are presented in the figure.
Images in Fig 2 were increased in size, re-ordered according to the discussion order in main text. Color schemes were described in legend.
6. Fig. 3A: The numbering on the legend is too small to see clearly.
Fig 3A was modified to increase the size for the numbering in the legend.
7. Fig. 3D&E: Labels or color descriptions should be included in the figure or legend as well as asterisks or arrows to indicate sites referenced in the text.
Fig 3 was modified to include labels for different structural domains. Color schemes were included in figure legend.
8. Fig. 5: It could be useful to include antibodies mentioned in the text under their respective mechanisms in this figure.
Name of mAbs demonstrated with indicated antiviral mechanisms were included in modified Fig 5.
9. Lines 239-240: “infection by mAbs” is confusing. The sentence should be reworded, such as “Mounting evidence suggests that Fc-effector functions elicited by mAbs play a critical role in protection against infection.”
In page 15, line 10-12, the sentence was changed to “Mounting evidence suggests that Fc-effector functions elicited by mAbs play a critical role in protection against infection”.
Reviewer 3 Report
In this review, the authors provide a concise summary of the antiviral activities of CHIKV monoclonal antibodies.
One suggestion I have is to provide more information within the figure legends so the reader can better understand the figures.
For example:
Figure 1D: The authors reference the domains of E2 and E1 in a majority of the text, yet the text is small and hard to read.
Figure 3: More detail is needed in the figure legend. The authors should mention what each color signifies in 3b and 3D/E. Also there is no reference to the dotted circle in Figure 3D/E. The antibody should also be laced in Figure 3E.
Figure 5: More information is needed in this legend. What do the abbreviations stand for? Also, a brief description of the model would be helpful. There is very little text in the main body of the review.
Author Response
In this review, the authors provide a concise summary of the antiviral activities of CHIKV monoclonal antibodies.
One suggestion I have is to provide more information within the figure legends so the reader can better understand the figures.
For example:
Figure 1D: The authors reference the domains of E2 and E1 in a majority of the text, yet the text is small and hard to read.
Fig. 1D was reproduced with larger labels.
Figure 3: More detail is needed in the figure legend. The authors should mention what each color signifies in 3b and 3D/E. Also there is no reference to the dotted circle in Figure 3D/E. The antibody should also be laced in Figure 3E.
Fig 3 was modified to include labels for different structural domains. Color schemes were included in figure legend.
Figure 5: More information is needed in this legend. What do the abbreviations stand for? Also, a brief description of the model would be helpful. There is very little text in the main body of the review.
Legend for Fig 5 was revised to include description for the models and full names for the abbreviations.